# Elucidating Differences in Early-Stage Centrosome Amplification in Primary and Immortalized Mouse Cells

**DOI:** 10.3390/ijms25010383

**Published:** 2023-12-27

**Authors:** Masakazu Tanaka, Masaki Yamada, Masatoshi Mushiake, Masataka Tsuda, Masanao Miwa

**Affiliations:** 1Division of Neuroimmunology, Joint Research Center for Human Retrovirus Infection, Kagoshima University, 8-35-1 Sakuragaoka, Kagoshima 890-8544, Japan; 2Faculty of Bioscience, Nagahama Institute of Bio-Science and Technology, 1266 Tamura, Nagahama 526-0829, Japanm_miwa@nagahama-i-bio.ac.jp (M.M.)

**Keywords:** centrosome amplification, chromosome instability, cell differentiation, PARP inhibition, DNA damage (doxorubicin, irradiation)

## Abstract

The centrosome is involved in cytoplasmic microtubule organization during interphase and in mitotic spindle assembly during cell division. Centrosome amplification (abnormal proliferation of centrosome number) has been observed in several types of cancer and in precancerous conditions. Therefore, it is important to elucidate the mechanism of centrosome amplification in order to understand the early stage of carcinogenesis. Primary cells could be used to better understand the early stage of carcinogenesis rather than immortalized cells, which tend to have various genetic and epigenetic changes. Previously, we demonstrated that a poly(ADP-ribose) polymerase (PARP) inhibitor, 3-aminobenzamide (3AB), which is known to be nontoxic and nonmutagenic, could induce centrosome amplification and chromosomal aneuploidy in CHO-K1 cells. In this study, we compared primary mouse embryonic fibroblasts (MEF) and immortalized MEF using 3AB. Although centrosome amplification was induced with 3AB treatment in immortalized MEF, a more potent PARP inhibitor, AG14361, was required for primary MEF. However, after centrosome amplification, neither 3AB in immortalized MEF nor AG14361 in primary MEF caused chromosomal aneuploidy, suggesting that further genetic and/or epigenetic change(s) are required to exhibit aneuploidy. The DNA-damaging agents doxorubicin and γ-irradiation can cause cancer and centrosome amplification in experimental animals. Although doxorubicin and γ-irradiation induced centrosome amplification and led to decreased p27Kip protein levels in immortalized MEF and primary MEF, the phosphorylation ratio of nucleophosmin (Thr199) increased in immortalized MEF, whereas it decreased in primary MEF. These results suggest that there exists a yet unidentified pathway, different from the nucleophosmin phosphorylation pathway, which can cause centrosome amplification in primary MEF.

## 1. Introduction

In a normal animal cell, one centrosome is present in the G1 phase of the cell cycle. Duplication of centrosome and DNA occurs in a coordinated manner in animal cells. The centrosome acts as a microtubule-organizing center during cell division and is an intracellular organelle essential for accurate chromosome partitioning [1,2]. In general, one or two centrosomes are present in cells; however, they may over-duplicate, a phenomenon known as centrosome amplification, which can be attributed to various causes [3,4]. Centrosome amplification occurs due to not only endogenous genetic instability but also exogenous irradiation and chemical agents [5]. Although it has been proposed earlier that anomalies in the number of centrosomes lead to chromosome instability and cause cancer [6], the precise mechanism underlying this event remains unclear.

In several cancer cells, genetic instability is associated with a deficiency of functions in DNA damage response genes, including mutation or deletion of *TP53*, or DNA damage response kinase genes such as *ATM* and *ATR* [7]. DNA double-strand breaks not only cause fragmentation of chromosomes but also occasionally cause chromosome recombination through the DNA repair mechanism mistakenly joining the cut ends of different chromosomes [8]. Such DNA damage is monitored by DNA repair mechanisms, including ATM or ATR protein kinase, and inhibits cell cycle progression [9]. However, when accumulating DNA damage in the cells cannot be repaired, apoptosis then occurs.

The p27Kip protein has been identified as one of the important tumor suppressor proteins together with the p53 protein, which regulates cell cycle progression. Studies have demonstrated that mice lacking one of the alleles of *p27* are highly susceptible to carcinogenesis induced by X-ray irradiation or mutagen administration [10,11]. When the *p27* gene is knocked out in mice, normal development of embryos occurs, but due to excessive cell proliferation, the body weight is approximately 30–50% larger than that of wild-type mice [12]. Although *p27* gene mutation is hardly recognized in cancers, p27 degrading activity was found to accelerate in cancers with poor prognosis. p27 is a target protein of Skp2 for degradation. Studies have reported that *Skp2* knockout mice exhibited abnormal accumulation of p27, polyploidy, centrosome amplification, and increased apoptosis [13,14].

We previously reported that the presence of a novel pathway for centrosome amplification involves post-translational modifications such as poly ADP-ribosylation that does not require DNA lesions [15]. We also reported that the inhibitor of poly(ADP-ribose) polymerase (PARP) induced centrosome amplification and chromosomal aneuploidy in CHO cells [16]. Poly(ADP-ribosyl)ation is a post-translational modification that involves the addition and polymerization of ADP-ribose residues to specific amino acid residues using NAD (nicotinamide adenine dinucleotide) as a substrate and is involved in various biological events, including genome stability and transcriptional regulation of genes [17]. PARP inhibition interferes with base excision repair, which is responsible for repairing single-strand breaks occurring 20,000 times in a normal cell in a day [18]. If, however, in cancer cells, which are defective in homologous recombination repair, the double-strand break, occurring at the replication fork, cannot be repaired, cell death occurs through synthetic lethality [19,20]. Actually, the PARP inhibitors are currently used to treat some cancers including ovary and breast [21].

Therefore, we hypothesized that centrosome amplification is one of the earliest changes occurring during carcinogenesis, which, in turn, leads to cancer with chromosomal aneuploidy. Since immortalized cells undergo several secondary changes during long-time culture, we used primary cultured cells in addition to immortalized cells to compare the response in centrosome amplification. We found that while the post-immortalized cells were highly sensitive to the induction of aberrant centrosome numbers by PARP inhibitors, the primary cells were less sensitive, and further genetic and/or epigenetic events are required to induce aneuploidy. Therefore, the present study was conducted to elucidate the mechanism of DNA damage-induced centrosome amplification using primary cultured cells. In particular, we propose that the findings obtained using primary cultured cells would provide important information regarding the initial stage of carcinogenesis and the signal transduction pathways that cause centrosome amplification with or without DNA-damaging agents.

## 2. Results

### 2.1. Evaluation of Centrosome Amplification after Treatment with PARP Inhibitors in Immortalized MEF and Primary MEF

Using CHO-K1 cells, we previously reported that centrosome amplification induced by the PARP inhibitor 3-aminobenzamide (3AB) does not necessarily require DNA damage [15].

In this study, we examined whether there were any differences in the centrosome amplification mechanism between mouse immortalized cells that had already been immortalized (immortalized MEF) and mouse primary cultures prepared directly from living tissue (primary MEF). We then observed that cell proliferation was significantly decreased 72 h after 3AB treatment in immortalized MEF and primary MEF. When the cell cycle pattern was analyzed using flow cytometry, neither immortalized MEF nor primary MEF were found to differ with or without 3AB treatment (Figure 1A).

However, abnormal centrosome amplification was detected in immortalized MEF but not in primary MEF 72 h after 3AB treatment, which did not lead to DNA damage (Figure 1B). We investigated whether primary MEF retain a state similar to that in vivo and whether 3AB treatment is insufficient because of the high homeostatic function of primary MEF. Therefore, we used AG14361, which has been identified to be a more potent and specific inhibitor of PARP1 and PARP2 [22]. When primary MEF were incubated with 5 µM AG14361 for 72 h, there was a significant increase in the number of cells containing three or more spots of γ-tubulin, a marker of the centrosome (Figure 2A). Moreover, the majority of spots of γ-tubulin colocalized with the spots of centriole, thus confirming that the actual number of centrosomes in primary MEF was increased via AG14361 treatment (Figure 2B, Table 1). The proliferation of cells was significantly reduced at 72 h after incubation with 5 µM AG14361 without significant changes in the flow cytometric patterns (Figure 2C), which was similar to what has been observed with 7 mM 3AB treatment (Figure 1A).

### 2.2. Examination of Chromosome Number Abnormality after PARP Inhibition

In addition to the numerical amplification of centrosomes (centrosome amplification), cancer cells often exhibit changes in the number of chromosomes (aneuploidy) and multipolar spindle formation [23,24]. In primary MEF, the normal chromosome number was 2*n* = 40 ± 4 and 80 ± 4 (Figure 2D). However, aneuploidy was not observed with 7 mM 3AB or 5 µM AG14361 treatment in primary MEF (Figure 2D). Meanwhile, in immortalized MEF, although centrosome amplification was detected (Figure 1B) [15], aneuploidy was not observed (Figure 3A,B).

### 2.3. Exploring the Pathway of Centrosome Amplification by Doxorubicin and γ-Irradiation in Primary MEF

Immortalized cells have been determined to constitute a group of cells formed by immortalization, and their properties, such as the length of telomeres, differ from those of primary cells [25]. Studies have also reported that susceptibility to drugs and ionizing radiation depends on cell characteristics [26,27]. Therefore, we examined the signal transduction pathway of centrosome amplification in primary MEF through doxorubicin and γ-ray irradiation that can cause DNA double-strand breakages.

The cell viabilities after treatment with doxorubicin and γ-irradiation for 48 h were 62% and 51%, respectively, compared to those of control (Figure 4A). The image showing centrosome amplification is depicted in Figure 4B. As it has been reported that a failure of cytokinesis during cell division is a cause of centrosome amplification [28], we then analyzed the flow cytometric patterns of primary MEF. However, we seldom detected any change in the DNA histogram patterns (Figure 4C), which probably indicated that centrosome amplification was not caused by the failure of cytokinesis under this condition and further suggested that it was caused by abnormal duplication of centrosomes in primary MEF.

To investigate the signal transduction pathway of centrosome duplication caused by the DNA-damaging agents, we examined the changes in the protein levels of p53, p21, Skp2, and p27. Moreover, we analyzed nucleophosmin (NPM), which serves as a regulator of centrosome duplication [29]. We then explored the effect of doxorubicin as a DNA-damaging agent on primary MEF (Figure 4D, Table 2, Appendix A). Although no significant change in the p53 protein level was detected (*p* = 0.18), the p21 protein level was observed to have increased by 2.2-fold (*p* < 0.05). In addition, the p27 level was decreased to 0.3-fold (*p* < 0.01), associated with an increase in the Skp2 protein level by 2.4-fold (*p* < 0.05). The protein level of NPM phosphorylation at the Thr199 (pNPM) protein level was reduced to 0.7-fold (*p* < 0.05), with no significant change in NPM level (*p* = 0.36), and, consequently, the ratio of pNPM/NPM was reduced to 0.77 (*p* = 0.01).

We then investigated the effect of γ-irradiation. We detected an increase of 1.5-fold in p53 (*p* < 0.05) and 1.3-fold in p21 (*p* = 0.07) protein levels, whereas the p27 protein level was found to be only slightly reduced to 0.9-fold, with no significant change in the Skp2 level (*p* = 0.19). Therefore, the increase in p53 and p21 protein levels is a dominant phenomenon. Interestingly, although both NPM and pNPM protein levels were increased by 1.4- and 1.9-fold, respectively (*p* < 0.05), the ratio of pNPM/NPM was decreased to 0.75 (*p* < 0.01) (Table 2, Appendix A).

To determine whether the Skp2-p27 axis plays a key role in the suppression of centrosome amplification in primary MEF after DNA damage, we analyzed the primary MEF derived from wild-type, *Skp2*^−/−^, and *p27*^−/−^ mice (Figure 4E). First, doxorubicin treatment was determined to significantly increase centrosome amplification by 2.9-fold in wild-type primary MEF (*p* < 0.01), 5.1-fold in *Skp2*^−/−^ primary MEF (*p* < 0.01), and 5.9-fold in *p27*^−/−^ primary MEF (*p* < 0.01) compared to that in each group before treatment. Then, among the three types of primary MEF, those derived from *Skp2*^−/−^ and *p27*^−/−^ mice exhibited further increases in centrosome amplification significantly by 1.4- and 1.5-fold, respectively (*p* < 0.01 and *p* < 0.01, respectively), compared to that in wild-type primary MEF. Second, treatment with γ-irradiation increased centrosome amplification by 4.8-fold in primary MEF (*p* < 0.01), 6-fold in *Skp2*^−/−^ primary MEF (*p* < 0.01), and 5.6-fold in p27^−/−^ primary MEF (*p* < 0.01) compared to that in wild-type primary MEF. No significant difference in centrosome amplification after γ-irradiation was observed among wild-type primary MEF, *Skp2*^−/−^ primary MEF, and *p27*^−/−^ primary MEF (Figure 4E).

### 2.4. Exploring the Pathway of Centrosome Amplification by Doxorubicin and γ-Irradiation in Immortalized MEF

First, we analyzed the cell cycle patterns of immortalized MEF after treatment with doxorubicin and γ-irradiation. At 24 h after treatment with doxorubicin and γ-irradiation, polyploid cells (>4N) were detected with an increase in cell population in the G2/M phase (Figure 4F) in complete difference to that in primary MEF (Figure 4C). Therefore, some portion of centrosome amplification, found in immortalized MEF, might be caused due to errors in cytokinesis in addition to over-duplication of centrosomes.

Next, Western blot analyses were conducted to investigate the factors that cause centrosome amplification after treatment with the DNA-damaging agents doxorubicin and γ-irradiation (Figure 4G, Table 2, Appendix A). Doxorubicin treatment increased the p53 protein level by 1.4-fold (*p* < 0.05) and the p21 protein level by 1.5-fold (*p* = 0.06). The level of p27 was decreased to 0.5-fold (*p* < 0.05) associated with an increase in the protein level of Skp2 by 4.6-fold (*p* < 0.05). In contrast to primary MEF, the pNPM protein level was observed to have increased by 4.5-fold (*p* < 0.05) with no change in NPM level (*p* = 0.34), and the ratio of pNPM/NPM was increased by 4.4-fold (*p* = 0.01).

Treatment with γ-irradiation increased the p53 level by 1.4-fold (*p* < 0.05), but the p21 protein level was not increased (*p* = 0.27). Moreover, the p27 protein level was reduced to 0.3-fold (*p* < 0.01), associated with a 2.1-fold increase in the Skp2 level (*p* < 0.05). Similar to the effects of doxorubicin treatment, the pNPM protein level was observed to have increased by 2.1-fold (*p* < 0.05), and the ratio of pNPM/NPM was increased by 2.7-fold (*p* < 0.05) (Table 2, Appendix A). These results indicate that the p53/Skp2-p27-pNPM axis might be the common pathway for the effects of doxorubicin and γ-irradiation treatment in immortalized MEF.

## 3. Discussion

In experimental animal models, it has been observed that the initial in vivo responsiveness to DNA-damaging carcinogens and the repair capacity of cells determine the susceptibility to cancer [30]. The most important difference in the upstream signal transduction pathway of centrosome amplification by doxorubicin is that the Skp2-p27 pathway has been dominantly used in primary MEF, whereas both Skp2-p27 and p53-p21 pathways are used in immortalized MEF (Figure 4D,G,H). These results caused by doxorubicin treatment might be consistent with the findings that, in *Skp2*^−/−^ primary MEF or *p27*^−/−^ primary MEF, the centrosome amplification was further increased significantly (Figure 4E). Therefore, it is possible that the Skp2-p27 pathway is the initial pathway in evolution for the response to doxorubicin and the p53-p21 pathway was later used after cells were immortalized to become immortalized MEF. Suppression of centrosome amplification by p27 has been previously reported [31]. Overexpression of Skp2 has been observed in various human cancers associated with reduced survival and is considered to have oncogenic activity [32]. The importance of the Skp2 pathway was proposed by Davidovich et al., who reported that Skp2 is a predictor of response to doxorubicin-based chemotherapy in breast cancer [33].

In terms of γ-irradiation-induced centrosome amplification, the p53-p21 pathway, a canonical pathway by DNA-damaging agents, might be initially used in primary MEF, and the Skp2-p27 axis might also be used in immortalized MEF during the course of immortalization, where adjustments to various environmental stresses had been required. This could be deemed possible, because the Skp2-p27 pathway was originally present for centrosome amplification by doxorubicin in primary MEF as shown (Figure 4D,G). Hence, it is possible that the pathways of cell signaling in response to DNA-damaging agents differ due to the types of DNA-damaging agents and also the stage of cell differentiation, because in colorectal cancer, it was observed that the extent of Skp2-p21/p27 degradation by β-catenin or ubiquitination depends on the degree of cancer cell differentiation [34].

Furthermore, in primary MEF, it appears to be important that in the downstream pathway, the pNPM/NPM pathway is bypassed for centrosome amplification by both doxorubicin and γ-irradiation, whereas the pNPM/NPM pathway is commonly used in immortalized MEF (Figure 4H, Table 2, Appendix A). NPM has been established as one of the phosphorylation targets of cdk2/cyclin E that triggers centrosome duplication. Okuda et al. reported that anti-NPM antibodies, when microinjected into cells, blocked the phosphorylation of NPM and suppressed the initiation of centrosome duplication. Moreover, the expression of the nonphosphorylatable mutant of NPM in cells blocked centrosome duplication [29]. Therefore, it is possible that the pNPM/NPM pathway was responsible for centrosome amplification in immortalized MEF during or after immortalization in immortalized MEF, while a different pathway was utilized by primary MEF.

We previously reported that a PARP inhibitor, 3AB, was able to induce numerical centrosome amplification in immortalized MEF [35], which we confirmed in the present study. However, 3AB could not induce centrosome amplification, and a more potent and specific PARP inhibitor was required for primary MEF (Figure 1B and Figure 2A). Treatment with 3AB caused both centrosome amplification and chromosomal aberrations only in CHO-K1 cells, but no aneuploidy was observed in immortalized MEF and primary MEF, which might have certain stronger cellular mechanisms, e.g., suppressing c-Myc activation and maintaining the homeostasis of chromosomes in vivo (Table 3). In CHO-K1 cells, centrosome amplification was caused by PARP inhibition, and it was directly linked to an abnormal chromosome number, probably related to the p53 status of CHO-K1 cells, having a mutation in codon 211 in exon 6, resulting in a change from Thr (ACA) to Lys (AAA) [36].

The results of our study strongly suggest that the mechanisms underlying centrosome amplification with and without DNA damage are clearly different between immortalized MEF and primary MEF (Table 3). Thus, further analyses on the centrosomal regulatory mechanisms that control the stability of the genome of the primary cells should be conducted. In addition, we suggest exploring the signal transduction pathways using more specific PARP inhibitors to induce centrosome amplification in primary cultured cells would provide important information for clarifying the mechanism underlying the early stage of carcinogenesis and also for developing new therapeutics to treat cancer.

## 4. Materials and Methods

### 4.1. Isolation of Mouse Embryonic Fibroblasts (MEF)

C57BL/6J wild-type mouse embryos were isolated at the embryonic d12-14 (E12–14) stage; then, they were placed in ice-cold phosphate-buffered saline (PBS). The experimental protocol using mice was approved by the Institutional Animal Care and Use Committee of Kansai Medical University in compliance with the Guide for the Care and Use of Laboratory Animals. Embryo bodies of mice were minced after removing their heads, feet, and intestines and then digested for 20 min with 0.25% trypsin-EDTA at room temperature. The incubation with trypsin-EDTA was stopped by adding DMEM supplemented with 10% heat-inactivated fetal bovine serum, 100 U/mL penicillin, and 100 µg/mL streptomycin and then incubated at 37 °C in an atmosphere containing 5% CO_2_ and 100% humidity. The medium was replaced every 2 days. MEF were frozen at Passage 1. In this study, MEF were used within Passage 6.

### 4.2. Cells 

MEF were cultured as described previously [15]. All media were supplemented with 10% fetal bovine serum, 100 U/mL penicillin, and 100 µg/mL streptomycin. Cells were maintained at 37 °C in an atmosphere containing 5% CO_2_ and 100% humidity.

### 4.3. Reagents and Antibodies

Doxorubicin, used as a DNA-damaging agent, and 3AB, used as a PARP inhibitor, were purchased from Sigma and Tokyo Chemical Industry Co. (Tokyo, Japan), respectively. AG14361 was a kind gift from Professor Nicola Curtin (Newcastle University, Newcastle Upon Tyne, UK) [22]. Mouse monoclonal antibodies against KIP1/p27 (BD), mouse monoclonal antibodies against α-tubulin (Sigma, St. Louis, MO, USA), and rabbit polyclonal antibodies against p53 (SANTA CRUZ, Dallas, TX, USA), p21 (SANTA CRUZ), and Skp2 (H-435, SANTA CRUZ) were used for immunoblot analysis. As secondary antibodies, goat anti-mouse IgG HRP conjugate (Nacalai, Kyoto, Japan) and horse anti-rabbit IgG HRP conjugate (Cell Signaling, Danvers, MA, USA) were used.

### 4.4. Growth Inhibition Assays

Cells were seeded in 96-well plates (Thermo Fisher Scientific, Waltham, MA, USA); then, they were allowed to adhere overnight before exposure to γ-irradiation or doxorubicin. Growth inhibition was determined using the XTT cell proliferation assay II (Roche, Basel, Switzwerland) according to the manufacturer’s instructions. Absorbance was measured at 492 nm with a reference wavelength of 620 nm. All experiments were performed in triplicate.

### 4.5. Flow Cytometry

Cells were treated with γ-ray irradiation or doxorubicin and then incubated with PBS containing 0.1% Triton X-100, 0.5% RNase A, and 40 µg/mL propidium iodide. Flow cytometric analysis was conducted using FACSCalibur ver.2.1 and CellQuest software ver.3.1 (Becton Dickinson, Franklin Lakes, NJ, USA).

### 4.6. Indirect Immunofluorescence

Cells grown on coverslips were treated with AG14361, doxorubicin, and γ-irradiation for 72 h; fixed with 3.7% formalin for 10 min at room temperature and 100% methanol for 10 min at −20 °C; washed with PBS; and then permeabilized with 1% Triton X-100 in PBS for 5 min. Next, the cells were incubated using a blocking solution (5% fetal bovine serum in PBS) for 30 min and subjected to immunostaining. For co-immunostaining of γ-tubulin and α-tubulin, cells were probed using rabbit anti-γ-tubulin (1:200; Sigma) and mouse anti-α-tubulin (1:200; Sigma) antibodies for 1 h at room temperature. For centrin and γ-tubulin co-immunostaining, cells were probed with rabbit anticentrin (1:300; Sigma) and mouse anti-γ-tubulin (1:300; Sigma) antibodies for 1 h at room temperature. The antibody–antigen complexes were detected by incubating for 1 h with Alexa 488-conjugated goat anti-rabbit IgG (1:2500; Invitrogen, Waltham, MA, USA) or Alexa 594-conjugated goat anti-mouse IgG (1:2500; Invitrogen). Finally, the cells were counterstained with 4′,6′-diamidino-2-phenylindole (DAPI; Invitrogen).

### 4.7. Counting of Metaphase Spread Chromosomes

Cells were incubated in the presence of colcemid (0.5 µg/mL) for 6 h to enrich mitotic cells. The medium containing floating mitotic cells was saved. The remaining cells were trypsinized and pelleted together with the saved medium through centrifugation. The cell pellet was then gently resuspended in a hypotonic solution (65 mM KCl) and allowed to stand for 20 min at 37 °C. Next, the hypotonic solution was removed, after which a methanol acetic acid fixative was added, and the cells were allowed to stand for 3 min. The old fixative was later discarded, and then a fresh fixative was added. This procedure was repeated twice. A few drops of the suspension on coverslips were subjected to Giemsa staining and were then examined under a light microscope.

### 4.8. Fluorescence In Situ Hybridization (FISH) Analysis

FISH analyses were performed using chromosome painting probes that were purchased from Applied Spectral Imaging Ltd., Carlsbad, CA, USA (Whole chromosome painting, Tokyo Instruments Inc., Tokyo, Japan) and used according to the manufacturer’s protocols.

### 4.9. γ-Irradiation

Cells were exposed to 2- or 10-Gy irradiation at room temperature using a ^137^Cs source (Gammacell 40 Exactor; Nordion International, Ottawa, ON, Canada).

### 4.10. Western Blot Analysis

Cells were lysed using a lysis buffer (20 mM Tris-HCl (pH 7.5), 0.25% sodium deoxycholate, 0.025% sodium dodecyl sulfate (SDS), 150 mM NaCl, 5 mM EDTA, 1 mM NaF, 1 mM NaVO_4_, and 1% NP40), which contains protease inhibitors (EDTA-free protease inhibitor cocktail; Roche Diagnostics, Basel, Switzerland). The resulting cell lysates were incubated on ice for 30 min and then centrifuged at 20,000× *g* for 15 min at 4 °C. The proteins present in the supernatant were then denatured in a sample buffer (62.5 mM Tris-HCl (pH 6.8), 10% (*v*/*v*) glycerol, 2% SDS, 5% 2-mercaptoethanol, and 0.001% bromophenol blue). The protein samples were then separated using SDS-PAGE, and then they were transferred onto an Immobilon P-membrane (Millipore, Burlington, MA, USA). The membrane was incubated overnight with primary antibody at 4 °C and then with horseradish peroxidase-conjugated secondary antibody in PBS containing 5% skim milk and 0.05% (*v*/*v*) Tween 20 for 60 min at room temperature. Next, the membrane was washed with 0.05% Tween 20 in PBS. The antibody–antigen complex was visualized using enhanced chemiluminescence (Amersham Pharmacia, Chalfont, UK) according to the manufacturer’s protocol.

### 4.11. Statistical Analysis

After conducting an equality test of two variances, Student’s or Welch’s *t*-test with equal or unequal variances, respectively, was used in detecting differences between mean scores for the treatment groups. Meanwhile, Fisher’s exact test was used to detect differences between the incidence rates of centrosome amplification.

## Figures and Tables

**Figure 1 ijms-25-00383-f001:**
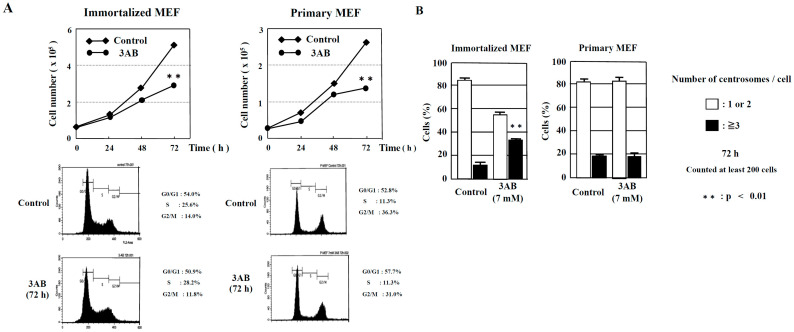
Numerical amplification of centrosomes by PARP inhibition is different between immortalized MEF and primary MEF. (**A**) Kinetic analyses of cell proliferation and flow cytometric analyses of immortalized MEF and primary MEF. MEF were treated with 3AB, trypsinized, and suspended in 0.4% trypan blue. Cells that did not stain with trypan blue were counted on a hemocytometer. ** *p* < 0.01 versus control with Student’s *t*-test. (**B**) Treatment with 3AB for 72 h induced centrosome amplification in immortalized MEF but not in primary MEF. More than 200 cells were examined, and the number of γ-tubulin spots per cell was measured. ** *p* < 0.01 versus control (Fisher’s exact test).

**Figure 2 ijms-25-00383-f002:**
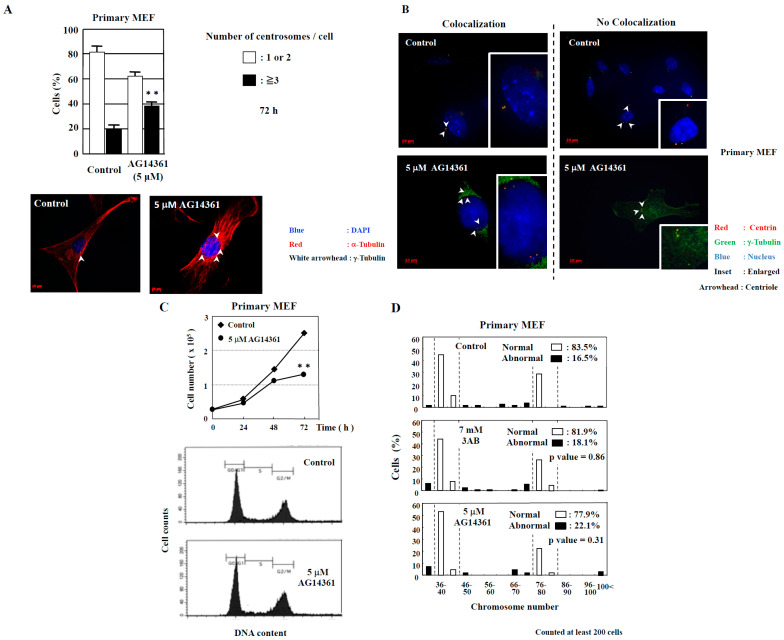
Centrosome amplification was induced in primary MEF using AG14361, a potent and specific PARP inhibitor, but abnormal number of chromosomes was not induced. (**A**) Increase in the number of centrosomes induced by AG14361 treatment for 72 h. Number of centrosomes per cell was counted for at least 200 cells. The lower two panels show the merged images of staining for DNA (blue), α-tubulin (red), and γ-tubulin (white arrowhead). ** *p* < 0.01 versus control with Student’s *t*-test. (**B**) Localization of centrioles in primary MEF. Centrioles are shown with centrin (red) and with white arrowheads, pericentriolar material (PCM) with γ-tubulin (green), and nucleus with DAPI (blue). (**C**) Kinetic analysis of cell proliferation and flow cytometric analyses of primary MEF. Primary MEF treated with AG14361 were trypsinized and suspended in 0.4% trypan blue. Cells that did not stain with trypan blue were counted on a hemocytometer. ** *p* < 0.01 versus control with Student’s *t*-test. Lower panels show flow cytometric analysis of primary MEF. (**D**) Distribution of the chromosome number 72 h after 3AB or AG14361 treatment in primary MEF. The vertical axis indicates the percentage of cells, and the horizontal axis indicates the number of chromosomes. The chromosome number 2*n* = 40 ± 4 and 80 ± 4 was taken as normal in primary MEF. More than 200 cells were examined. *p*-value was determined using Fisher’s exact test.

**Figure 3 ijms-25-00383-f003:**
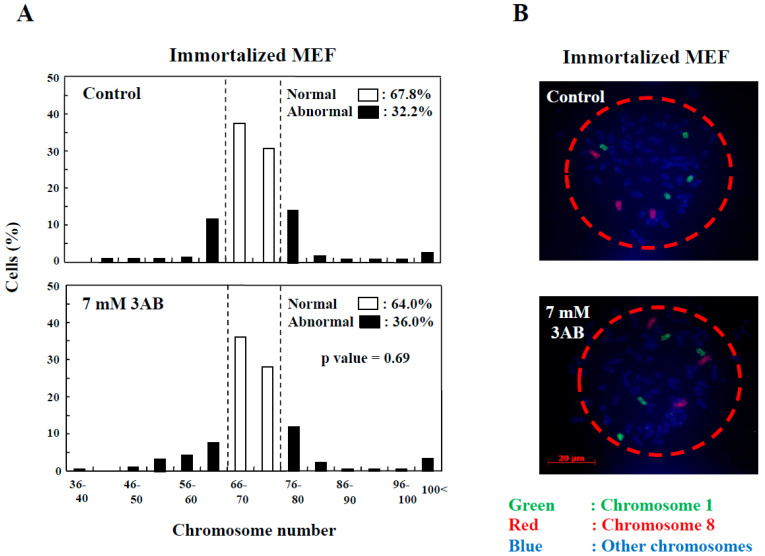
Centrosome abnormality and chromosomal instability after 3AB treatment were confirmed using other immortalized cells. (**A**) No significant change in the number of chromosomes 72 h after 3AB treatment in immortalized MEF. The number of chromosomes 2*n* = 70 ± 4 was taken as the normal number of chromosomes in immortalized MEF. The vertical axis indicates the percentage of cells, and the horizontal axis indicates the number of chromosomes. More than 200 cells were examined. *p*-value was determined using Fisher’s exact test. (**B**) Fluorescence in situ hybridization (FISH) images of a cell from immortalized MEF to depict no changes in the numbers of chromosome 1 and chromosome 8 per cell without or with 3AB treatment for 72 h. Nuclear area is surrounded by dashed red circle.

**Figure 4 ijms-25-00383-f004:**
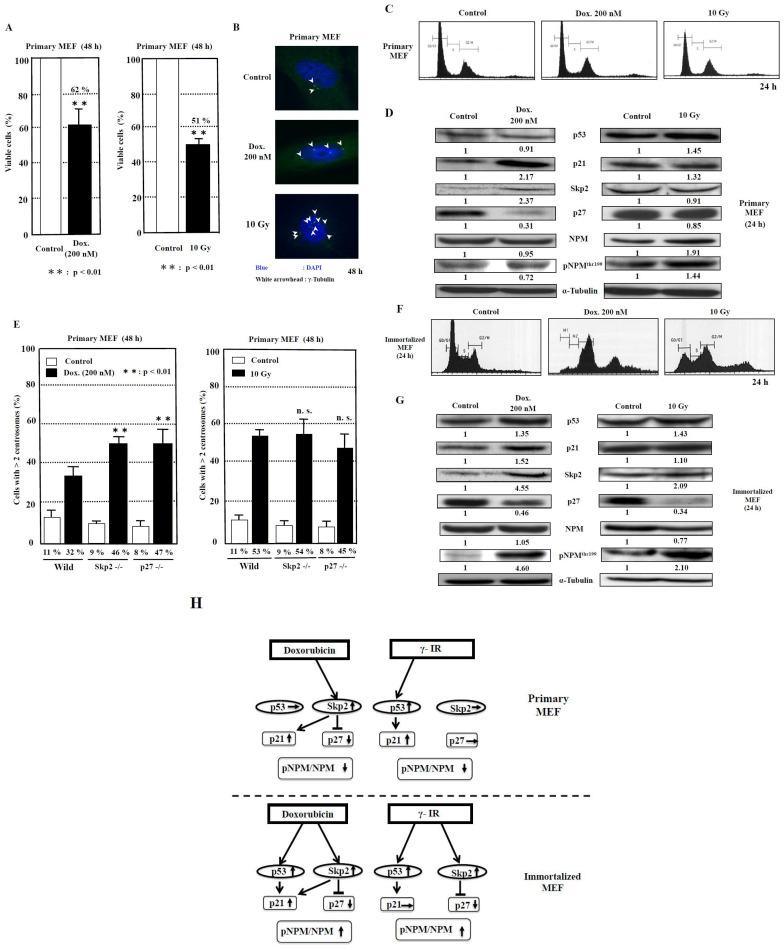
Analysis of centrosome amplification after DNA damage in primary MEF. (**A**) The measurement of viable cells after 48 h of doxorubicin (Dox.) treatment and 10-Gy γ-irradiation was conducted using XTT method. The control value is taken as 100%, and the standard deviations are shown with vertical bars. Statistical significance was determined using Student’s *t*-test, N = 5. (**B**) The images of centrosome amplification induced by doxorubicin and γ-irradiation treatment after 48 h. The arrowheads refer to the centrosomes. The nuclei were stained blue by Hoechst. (**C**) No significant change in the flow cytometric patterns of primary MEF 24 h after doxorubicin or γ-irradiation treatment. The vertical axis indicates the number of cells, and the horizontal axis indicates the DNA content of cells. (**D**) Changes in the levels of proteins related to centrosome duplication 24 h after 200 nM doxorubicin treatment and 10 Gy γ-irradiation to primary MEF. Antibodies were used to detect the respective proteins. Representative data from three independent experiments are presented. The density of each protein was calculated using ImageJ software ver.1.51 and was digitized. The density of each band was normalized to that of α-tubulin, and the ratio of the density of the respective protein treated with doxorubicin or γ-irradiation divided by that of untreated band is shown. The mean values from three experiments are described under the respective bands (Appendix A). (**E**) Centrosome amplification in wild-type, Skp2^−/−^, and p27^−/−^ primary MEF at 48 h after treatment with or without doxorubicin (Dox., 200 nM) or γ-irradiation (γ-IR, 10 Gy). Cells were immunostained with anti-γ-tubulin antibody, and the number of γ-tubulin spots was counted. More than 200 cells were examined. Columns, the mean of three independent experiments; bars, standard deviation. *p*-value was determined using Student’s *t*-test. **, *p* < 0.01, compared with wild-type primary MEF at 48 h after doxorubicin or γ-irradiation treatment. n.s., not significant. (**F**) Changes in the flow cytometric patterns of immortalized MEF 24 h after doxorubicin or γ-irradiation treatment. The vertical axis indicates the number of cells, and the horizontal axis indicates the DNA content of cells. (**G**) Changes in the levels of proteins related to centrosome duplication 24 h after 200 nM doxorubicin treatment and 10-Gy γ-irradiation to immortalized MEF, respectively. Antibodies were used to detect the respective proteins. Representative data from three independent experiments are presented. The density of each protein was calculated by ImageJ software ver. 1.51 and was digitized. The density of each band was normalized to that of α-tubulin, and the ratio of the density of the respective protein treated with doxorubicin or γ-irradiation was divided by that of untreated band. The mean values from three experiments are described under the respective bands (Appendix A). (**H**) The proposed model for the signal transduction pathways of centrosome amplification after DNA damage in primary MEF and immortalized MEF. The increase in density to over 1.2-fold and the decrease to below 0.8-fold of the control density in western blots are indicated by upward and downward arrows, respectively. The increase in the ratio of pNPM/NPM to over 1.2 and the decrease to below 0.8 are indicated by upward and downward arrows, respectively (Appendix A).

**Table 1 ijms-25-00383-t001:** Most of cells show colocalization of γ-tubulin and centrin spots.

Treatment	No. Cells with 1 or 2γ-Tubulin Spots	No. Cells with Colocalization (%)	No. Cells without Colocalization (%)	No. Cells with >2 γ- Tubulin Spots	No. Cells with Colocalization (%)	No. Cells without Colocalization (%)
Control	331	331 (100)	0 (0)	107	105 (98.1)	2 (1.9)
5 μM AG14361	296	293 (99)	3 (1)	156	152 (97.4)	4 (2.6)

Colocalization of γ-tubulin spots and centrin spots in primary MEF treated with or without the PARP inhibitor AG14361.

**Table 2 ijms-25-00383-t002:** Summary of changes in centrosome replication-related proteins 24 h after doxorubicin and γ-irradiation (γ-IR) treatment.

	DNA—Damaging Agents	Primary MEF	Cell Line MEF
p53	Dox.	No	+
	γ-IR	+	+
p21	Dox.	+	+
	γ-IR	+	No
Skp2	Dox.	+	+
	γ-IR	No	+
p27	Dox.	−	−
	γ-IR	No	−
NPM	Dox.	No	No
	γ-IR	+	−
pNPM (Thr 199)	Dox.	−	+
	γ-IR	+	+
pNPM/NPM	Dox.	+	+
	γ-IR	+	+
Centrosome amplification	Dox.	+	+
	γ-IR	+	+

The changes measured by ImageJ software were quantified, and those with no significant difference were defined as “No (no change)”. Changes in the levels of proteins related to centrosome duplication 24 h after 200 nM doxorubicin treatment and 10-Gy γ-irradiation to primary MEF and cell line MEF. The increase in density to over 1.2-fold of the control density in western blot is shown with “+” mark, and the decrease to below 0.8-fold of the control density is shown with “−” mark. Neither increase nor decrease of the density is shown with “No” change (Figure 4H, Appendix A).

**Table 3 ijms-25-00383-t003:** Response of primary MEF, cell line MEF, and CHO-K1 cells * to PARP inhibitors for centrosome amplification and chromosomal aneuploidy.

Cell	PARP Inhibitor	Centrosome Abnormality	Chromosomal Aneuploidy
Primary MEF	AG14361	+	−
	3AB	−	−
Cell line MEF	3AB	+	−
CHO–K1	3AB	+	+

* Results from the reference [16]. +: Present (Abnormal) −: Absent (Normal).

## Data Availability

Data is contained within the article and Appendix A.

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
