# Peer review of "Elucidating Differences in Early-Stage Centrosome Amplification in Primary and Immortalized Mouse Cells"

_ijms, 2023, doi:10.3390/ijms25010383_

Round 1
Reviewer 1 Report
Comments and Suggestions for Authors
In this article, the authors compared centrosome amplification in primary and immortalized mouse cells. They have found some difference between primary MEF and immortalized MEF in response to centrosome amplification inducers and DNA-damaging agents. Results in this article suggest that there exists a yet unidentified mechanism of centrosome amplification in primary MEF. The aim of this study is to get information regarding the initial stage of carcinogenesis. Which is meaningful to cancer biology study and anti-cancer drug development.
There are few questions:
1) In line 27, ", the phosphorylation rate of nucleophosmin (Thr199) to nucleophosmin" this sentence is confusing.
2) In figure 1, is there any difference in the cell proliferation rate between immortalized cells and primary cells? If the cell proliferation rate was different from each other, will that contribute to the difference in centrosome amplification?
3) Is the immortalized cell used to mimic the initial stage of carcinogenesis? What is the method to generate immortalized cell line? Have you considered the influence of immortalization processing on the experiment design and results?
4) Is that possible to conduct your study using induced carcinomas cells instead of immortalized cell?
Author Response
Comments by Reviewer 1
Comment 1)
1) In line 27, ", the phosphorylation rate of nucleophosmin (Thr199) to nucleophosmin" this sentence is confusing.
Response 1)
- Thank you for the comment. We changed the sentence “, the phosphorylation rate of nucleophosmin (Thr199) to nucleophosmin” to “, the phosphorylation ratio of nucleophosmin (Thr199)”. [Lines:56-57]
Comment 2)
- In figure 1, is there any difference in the cell proliferation rate between immortalized cells and primary cells? If the cell proliferation rate was different from each other, will that contribute to the difference in centrosome amplification?
Response 2)
Indeed, our results showed that cell proliferation was about two times faster in immortalized cells than in Primary cells. It is an interesting question. However, we have not analyzed this question in detail. We must analyze this question in the future work.
Comment 3)
3) Is the immortalized cell used to mimic the initial stage of carcinogenesis? What is the method to generate immortalized cell line? Have you considered the influence of immortalization processing on the experiment design and results?
Response 3)
This is a very important point. The immortalized MEF, we used, is spontaneously immortalized MEF. It might be possible that oncogene-transduced immortalized MEF might have different characteristics. This is an important question, which should be analyzed in the future experiment.
Comment 4)
4) Is that possible to conduct your study using induced carcinomas cells instead of immortalized cell?
Response 4)
We think that it is possible to use carcinoma cells induced by chemical carcinogen. This is also an interesting experiment for our future work.
Other change
To avoid confusion, caption of Fig. 4 was changed from “Centrosome abnormality and chromosomal instability after 3AB treatment were confirmed using other immortalized cells.” to
“Centrosome abnormality without chromosomal instability after 3AB treatment were confirmed using other immortalized cells.”

Reviewer 2 Report
Comments and Suggestions for Authors
In this manuscript entlited “Elucidating differences in early stage centrosome amplification in primary and immortalized mouse cells” Tanaka and collegues studied centrosome amplification in primary mouse embryonic fibroblasts (MEF) and immortalized MEF.
The topic of this study is very interesting since abnormal proliferation of centrosome number has been reported in several types of cancer and in precancerous condition. The Authors employed several different experimental approaches, and the experimental work is well orchestrated. So, in the opinion of this reviewer the manuscript is surely of interest and worthy of being published after minor revisions.
Here my specific comments.
Introduction
Lines 34-36: here the Authors should add that centrosome is involved in cytoplasmic microtubules organization during interphase and in mitotic spindle assembling during cell division.
Line 37: here the Authors should clarify that one centrosome is present in the cell at interphase and two centrosomes are present during mitosis (to organize the bipolar spindle involved in chromatids segregation at anaphase).
Lines 78-80: “Since immortalized cells undergo several secondary changes during long time culture. Hence, we used primary cultured cells in addition to immortalized cells to compare the response in centrosome amplification”. Perhaps these two sentences could be joined.
Materials and Methods
Indirect immunofluorescence
Here this reviewer need some experimental elucidation mainly on the fixation method for indirect immunoflorescence. In my experience methanol at −20°C is a very good fixative for centrosomal proteins and microtubules staining so the authors should explain why they used the formalin, togheter with methanol to fix the cells.
Related to this I must point out that in Fig 2A the microtubules and the gamma-tubulin staining are not visible, neither in the original images. These could be due to a fixation problem? I retain that the Authors should improve this picture.
About Fig. 2B this reviewer highlight that the fluorescence pictures are very clear and evident in the original images but not in the figures present in the manuscript. I hope this aspect can be improved.
Results
Lines 200-201: Concerning Fig 1B the Authors should clearly explain how the number of centrosomes was counted.
Line 240: I'm sorry but I am not able to find Fig 3C and 3D in Fig. 3!!!
Line 260: Here the Authors should indicate the centrosome marker employed in Fig 4B.
About Fig. 4B this Reviewer note that, as also previously observed, in the original images the spots corresponding to centrosomes are very clear and visible while in the paper the fluorescence is much less visible. I hope this aspect can be improved.
Author Response
Comments by Reviewer 2
Comment 1)
Introduction
Lines 34-36: here the Authors should add that centrosome is involved in cytoplasmic microtubules organization during interphase and in mitotic spindle assembling during cell division.
Response to comment 1)
Thank you for the comments. We added the sentences “The centrosome is involved in cytoplasmic microtubule organization during interphase and in mitotic spindle assembling during cell division” at the beginning of the abstract as suggested. [Lines:37-38]
Comment 2)
Line 37: here the Authors should clarify that one centrosome is present in the cell at interphase and two centrosomes are present during mitosis (to organize the bipolar spindle involved in chromatids segregation at anaphase).
Response to comment 2)
According to the reviewer’s comment, we added the following sentence at the beginning of introduction; “In a normal animal cell, one centrosome is present in G1 phase of the cell cycle.” [Line:74]
Comment 3)
Lines 78-80: “Since immortalized cells undergo several secondary changes during long time culture. Hence, we used primary cultured cells in addition to immortalized cells to compare the response in centrosome amplification”. Perhaps these two sentences could be joined.
Response to comment 3)
We joined the two sentences as follows. “Since immortalized cells undergo several secondary changes during long-time culture, we used primary cultured cells in addition to immortalized cells to compare the response in centrosome amplification.” [Lines:121-123]
Comment 4)
Materials and Methods
Indirect immunofluorescence
Here this reviewer need some experimental elucidation mainly on the fixation method for indirect immunoflorescence. In my experience methanol at −20°C is a very good fixative for centrosomal proteins and microtubules staining so the authors should explain why they used the formalin, togheter with methanol to fix the cells.
Response to comment 4)
Thank you. We used both methanol and formalin in the present study because methanol is more likely to attenuate fluorochromes, while formalin is better at fixing cellular structural mechanisms.
Comment 5)
Related to this I must point out that in Fig 2A the microtubules and the gamma-tubulin staining are not visible, neither in the original images. These could be due to a fixation problem? I retain that the Authors should improve this picture.
Response to comment 5)
Thank you, we have improved Picture.
Comment 6)
About Fig. 2B this reviewer highlight that the fluorescence pictures are very clear and evident in the original images but not in the figures present in the manuscript. I hope this aspect can be improved.
Response to comment 6)
Thank you, we have improved Picture.
Comment 7)
Results
Lines 200-201: Concerning Fig 1B the Authors should clearly explain how the number of centrosomes was counted.
Response to comment 7)
Thank you. I have rewritten the following. 
We changed the sentence “ 2.6. Indirect immunofluorescence.” to “2.6. Centrosome Amplification Judgement”.
Further added at the end of the sentence.
“The cells were sealed on glass slides (Fluoromount/Diagnostic BioSystems, Pleasanton, CA, USA), stored at 4 ℃ under light-shielded conditions overnight, and then observed using a fluorescence microscope (ZEISS FLV1000). The number of spots of γ-tubulin per cell was measured. Cells with more than 2 centrosomes were defined to have centrosome amplification. Each value represents the mean ± SD of 3 independent experiments.”
Comment 8)
Line 240: I'm sorry but I am not able to find Fig 3C and 3D in Fig. 3!!!
Response to comment 8)
We apologize our mistake. The data of CHO had been already published in Ref 16. Therefore, we deleted the sentence of “In contrast, in the CHO-K1 cells, which exhibited centrosome amplification upon incubation with 3AB (15), there was significant aneuploidy at 72 h after treatment with 7 mM 3AB (Figs. 3C, 3D).”
Also, to make clearer, we changed the sentences in abstract “In this study, we have compared primary mouse embryonic fibroblasts (MEF) and immortalized MEF using a poly(ADP-ribose) polymerase (PARP) inhibitor, 3-aminobenzamide (3AB), which is known to be nontoxic and nonmutagenic but could induce centrosome amplification in immortalized MEF. Although centrosome amplification was induced with 3AB treatment in immortalized MEF, a more potent PARP inhibitor, AG14361, was required for primary MEF. 3AB has been identified to induce centrosome amplification and chromosomal aneuploidy in CHO-K1 cells.” to
“Previously, we reported that a poly(ADP-ribose) polymerase (PARP) inhibitor, 3-aminobenzamide (3AB), which is known to be nontoxic and nonmutagenic but could induce centrosome amplification and chromosomal aneuploidy in CHO-K1 cells. In this study, we compared primary mouse embryonic fibroblasts (MEF) and immortalized MEF using 3AB. “Although centrosome amplification was induced with 3AB treatment in immortalized MEF, a more potent PARP inhibitor, AG14361, was required for primary MEF.” [Lines:44-50]
The description that “CHO-K1 cells and" was deleted in Materials and Methods accordingly.
Comment 9)
Line 260: Here the Authors should indicate the centrosome marker employed in Fig 4B.
Response to comment 9)
Thank you.
In Fig. 4B, we have marked "Blue: DAPI, White arrowhead: γ-Tubulin".
Comment 10)
About Fig. 4B this Reviewer note that, as also previously observed, in the original images the spots corresponding to centrosomes are very clear and visible while in the paper the fluorescence is much less visible. I hope this aspect can be improved.
Response to comment 10)
Thank you, we have improved Picture.
Other change
To avoid confusion, caption of Fig. 4 was changed from “Centrosome abnormality and chromosomal instability after 3AB treatment were confirmed using other immortalized cells.” to
“Centrosome abnormality without chromosomal instability after 3AB treatment were confirmed using other immortalized cells.”
